# Efficacy of Combination Therapy with Lenvatinib and Radioactive Iodine in Thyroid Cancer Preclinical Model

**DOI:** 10.3390/ijms23179872

**Published:** 2022-08-30

**Authors:** Kensuke Suzuki, Hiroshi Iwai, Keita Utsunomiya, Yumiko Kono, Tadashi Watabe, Yoshiki Kobayashi, Dan Van Bui, Shunsuke Sawada, Yasutaka Yun, Akitoshi Mitani, Kenta Fukui, Haruka Sakai, Hanh Hong Chu, Nguyen Manh Linh, Noboru Tanigawa, Akira Kanda

**Affiliations:** 1Department of Otolaryngology, Head and Neck Surgery, Kansai Medical University, Hirakata 573-1010, Osaka, Japan; 2Department of Radiology, Kansai Medical University, Hirakata 573-1010, Osaka, Japan; 3Department of Nuclear Medicine and Tracer Kinetics, Graduate School of Medicine, Osaka University, Suita 565-0871, Osaka, Japan

**Keywords:** thyroid cancer, lenvatinib, radioactive iodine, combination therapy, sodium iodide symporter

## Abstract

Patients with differentiated thyroid cancer (DTC) usually have good prognosis, while those with advanced disease have poor clinical outcomes. This study aimed to investigate the antitumor effects of combination therapy with lenvatinib and ^131^I (CTLI) using three different types of DTC cell lines with different profiling of sodium iodide symporter (NIS) status. The radioiodine accumulation study revealed a significantly increased radioiodine uptake in K1-NIS cells after lenvatinib treatment, while there was almost no uptake in K1 and FTC-133 cells. However, lenvatinib administration before radioiodine treatment decreased radioiodine uptake of K1-NIS xenograft tumor in the in vivo imaging study. CTLI synergistically inhibited colony formation and DTC cell migration, especially in K1-NIS cells. Finally, ^131^I treatment followed by lenvatinib administration significantly inhibited tumor growth of the NIS-expressing thyroid cancer xenograft model. These results provide important clinical implications for the combined therapy that lenvatinib should be administered after ^131^I treatment to maximize the treatment efficacy. Our synergistic treatment effects by CTLI suggested its effectiveness for RAI-avid thyroid cancer, which retains NIS function. This potential combination therapy suggests a powerful and tolerable new therapeutic strategy for advanced thyroid cancer.

## 1. Introduction

Thyroid cancer is the most common type of malignant endocrine tumor [1], with an increased incidence over the last four decades [2]. Differentiated thyroid cancer (DTC), which includes papillary and follicular types, accounts for the vast majority of thyroid cancers [3]. While 90% of patients with DTC survive for at least 10 years, those with advanced disease have poor clinical outcomes [4]. Patients with surgically unresectable advanced DTC have only two choices, radioactive iodine therapy (RAI; ^131^I) and tyrosine kinase inhibitors (TKIs). RAI therapy is based on the expression of sodium iodide symporter (NIS) at the plasma membrane of normal and tumoral thyroid epithelial cells [5] and is the gold standard for adjuvant therapy postoperatively or for the treatment of distant metastases from DTC [6,7]. However, two-thirds of patients with metastatic DTC are or will become RAI-refractory [8,9], which is correlated with a less differentiated tumor state and decreased NIS expression [5]. The median survival of RAI-refractory DTC is only 2.5–3.5 years from the diagnosis of distant metastases [10,11].

Recently, some novel TKIs have improved progression-free survival of patients with thyroid cancer [12,13]. Lenvatinib is a multi-targeted TKI that selectively inhibits vascular endothelial growth factor receptors 1–3, fibroblast growth factor receptors 1–4, platelet-derived growth factor receptor-alpha, KIT, and RET [14,15,16]. A phase III study of 392 patients with RAI-refractory DTC (SELECT trial) revealed that lenvatinib improved progression-free survival and response rate [13]. However, TKIs, including lenvatinib, have a high incidence of treatment-related adverse events, and most patients discontinue their use due to unresponsiveness [17]. Furthermore, TKIs are often reserved for RAI-refractory patients with rapid tumor progression or severe symptoms, according to current treatment guidelines [18,19].

Concurrent chemoradiation therapy has become one of the standard initial treatments for advanced-stage head and neck squamous cell carcinoma (HNSCC) in recent years [20,21]. Cetuximab is the first molecular-targeted agent for HNSCC treatment with improved treatment results in combination with radiotherapy in a phase III study [22]. In contrast, no established regimen and only few basic studies have been documented about irradiation combined with chemotherapy or molecular-targeted therapy in thyroid cancer.

Interestingly, some clinical studies have reported cases with rapid tumor regression after concurrent combination therapy of lenvatinib and external beam radiation therapy (EBRT) without any serious complications [23,24]. We have previously reported the synergistic effects of combination therapy with lenvatinib and EBRT in human DTC cells and xenograft models [25]. Therefore, we hypothesized that combined therapy with lenvatinib and RAI (^131^I) has synergistic and systemic antitumor effects against thyroid cancer. To the best of our knowledge, this study is the first to evaluate the treatment effects of combination therapy with lenvatinib and ^131^I (CTLI) against DTC through a series of in vitro and in vivo studies. This study aimed to investigate and compare the efficacy of CTLI against different types of thyroid cancer cells that have different profiles for NIS status.

## 2. Results

### 2.1. Increased Intracellular Uptake of Radioiodine through Sodium Iodide Symporter after Lenvatinib Treatment In Vitro

We performed a ^125^I accumulation study after lenvatinib treatment for thyroid cancer cells K1, FTC-133, and K1-NIS to investigate the effect of lenvatinib on intracellular radioiodine accumulation. We confirmed that K1-NIS cells strongly expressed NIS compared with K1 cells (Appendix A). There was no statistically significant difference in the K1-NIS proliferation compared with K1 cells, as shown in Appendix A. Cellular uptake analysis showed high ^125^I uptake in the K1-NIS cells but almost no uptake in the K1 and FTC-133 cells (Figure 1A,B, Appendix A), suggesting that radioiodine is transported into DTC cells through NIS. K1-NIS cell uptake was significantly increased after lenvatinib treatment compared with the control (Figure 1B). The CPMin/CPMout ratio attained a plateau at approximately 15 min after adding ^125^I in K1-NIS cells. The CPMin/CPMout ratio in lenvatinib treatment groups in K1-NIS cells dose-dependently increased, indicating increased intracellular ^125^I uptake after lenvatinib treatment in vitro.

### 2.2. In Vivo DTC Radioiodine Avidity after Lenvatinib Treatment

We performed ^123^I imaging analyses in K1-NIS xenograft mouse model following oral lenvatinib administration based on the in vitro experiments to visualize the influence of radioiodine intake following the lenvatinib treatment. Compared with the vehicle group, those treated with 10 mg/kg lenvatinib for 5 days showed a significantly decreased ^123^I uptake (radioiodine incorporation) into the tumor (Figure 2A,B,D), while the ^123^I uptake into the thyroid was not different between treatment group and vehicle control (Figure 2C). Similarly, the radioactivity of counted tumor using an automatic ɣ counter was significantly decreased in the group treated with lenvatinib (Figure 2E). Surprisingly, results of in vivo imaging study were against the results of ^125^I accumulation study in vitro. Notably, the tumor weight was not significantly different between the vehicle and treatment group (Figure 2F).

### 2.3. Inhibition of Thyroid Cancer Cell Growth and Migration Using Combination Therapy with ^131^I and Lenvatinib In Vitro

The CTLI anticancer activity on thyroid cancer cell lines K1 and K1-NIS was investigated in comparison to ^131^I or lenvatinib monotherapy by colony formation and wound-healing assays. CTLI significantly suppressed a colony-forming ability compared to the control, as shown in Figure 3A,B, and this suppression effect was more pronounced in K1-NIS cells than in K1 cells. Expectedly, CTLI caused a synergistic inhibitory effect on colony formation in DTC cells compared to ^131^I or lenvatinib monotherapy. The migration of K1 and K1-NIS cells after ^131^I and lenvatinib treatment was also investigated using a wound-healing assay. Results showed that CTLI significantly inhibited migration ability in both cell lines, and the inhibitory effect was tended to be higher in K1-NIS cells than in K1 cells (Figure 3C,D). These results suggest that the combined therapy synergistically inhibited proliferation and migration ability of thyroid cancer cells, especially NIS-expressing DTC cells in vitro.

### 2.4. Synergistic Anticancer Effects by Combination Therapy in Thyroid Cancer Xenografts

We then studied the antitumor effects of CTLI in vivo. We used tumor xenografts of K1-NIS cells murine model following monotherapy or combined therapy. Significant differences were found in tumor growth suppression between the combined therapy and the other groups, as shown in Figure 4A,C. Moreover, similar results were found with tumor weight (Figure 4D), indicating the synergistic effects of the combined therapy. Notably, body weight, an animal health indicator, was not significantly different among the treatment groups (Figure 4B), suggesting no emergence of toxicity effects by lenvatinib administration. Together, these results conclusively show the potent anticancer activity of CTLI in the DTC xenograft models.

## 3. Discussion

An established regimen of such combination therapy has not been documented in thyroid cancer, although radiation therapy in combination with chemotherapy or molecular-targeted agents is one of the gold standard therapies for HNSCC treatment [20,21]. Based on some clinical case reports that documented the durable response by combination therapy with lenvatinib and radiation [23,24], we hypothesized that this combination therapy could be applied for DTC treatment. We previously reported the antitumor effects of combination therapy with lenvatinib and EBRT in both human DTC cells and xenograft models [25]. This study replaced EBRT with ^131^I and demonstrated that the combination therapy with lenvatinib and ^131^I synergistically inhibited cell proliferation and migration ability in vitro, as well as tumor growth in the xenograft model. These findings suggest that this combination therapy could be a novel approach for patients with advanced thyroid cancer.

Recently, a growing understanding of molecular oncology has revealed the thyroid cancer pathogenesis [26]. One of the most common oncogenic changes observed in PTC is the *BRAF*V600E mutation, which reduces the NIS gene expression and leads to subsequent RAI-refractory metastatic disease [27]. Several authors reported that K1 (derived from PTC) cell line carries *BRAF*V600E mutation [16,28]. This study used three different types of thyroid cancer cell lines, K1, FTC-133, and K1-NIS, which were obtained by transfection using the human NIS gene to K1 cells [29] to investigate and compare the efficacy of CTLI against thyroid cancer cells with different profiling of NIS status.

The radioiodine accumulation study revealed a significantly increased radioiodine uptake in K1-NIS cells after lenvatinib treatment, while almost no uptake was shown in the K1 and FTC-133 cells. These results indicated that lenvatinib could potentially upregulate the function of radioiodine uptake in NIS-expressing thyroid cancer in vitro. Conversely, lenvatinib administration before ^123^I treatment decreased the radioiodine uptake of NIS-expressing thyroid cancer in vivo, suggesting that tumor angiogenesis inhibition of lenvatinib affects the radioiodine avidity. Lenvatinib was administered 3 h after treatment with ^131^I in vivo treatment study when it was expected to accumulate in the tumor based on the imaging study results. Hence, significant differences were observed in tumor growth of NIS-expressing thyroid cancer xenograft model between the combined therapy and the other groups. These results may provide important clinical implications for the combination therapy in which lenvatinib should be administered after ^131^I treatment to maximize treatment efficacy.

In vitro experiments demonstrated the inhibitory effects of CTLI on colony formation and migration of both K1 and K1-NIS. Furthermore, the inhibitory effects were more pronounced in NIS-expressing thyroid cancer (K1-NIS), which is estimated to be sensitive to ^131^I. In vitro and in vivo experiments on synergistic treatment effects by combined therapy suggested that it might be most effective for RAI-avid thyroid cancer, which retains NIS function. Clinically, the combined therapy could be applied as an alternative postoperative ablation treatment or adjuvant RAI for high-risk DTC and high-dose RAI for metastatic thyroid cancer. Supporting our results, Sheu et al. reported two patients with advanced DTC who were successfully rescued by lenvatinib treatment complementary with RAI in the durable response [30].

This study has some limitations. The iodine avidity of cancer cells should be considered for the potential application of combined therapy with lenvatinib and RAI. Enhancing iodine uptake into tumors is necessary to restore the sensitivity of RAI-refractory thyroid cancer to ^131^I [5]. Our in vivo experiments showed a synergistic treatment effect of lenvatinib on RAI but without increased iodine uptake into the tumor. The use of agents that enhance NIS function and RAI uptake in combination with CTLI may be necessary for the clinical application.

The loss of differentiated features and resistance to RAI in thyroid cancer correlates with the degree of mitogen-activated protein kinase (MAPK) activation [31]. Recently, several authors reported that some MAPK pathway inhibitors, namely BRAF inhibitors, such as dabrafenib [32], and MEK inhibitors, such as selumetinib [33], increased RAI uptake and improved the therapeutic effect of RAI, which is called the “redifferentiation” strategy [5]. The combination of these MAPK pathway inhibitors and lenvatinib could theoretically improve the clinical response to ^131^I treatment in DTC, especially in patients with RAI-refractory lesions. Further investigation is required for the clinical application of this combination therapy to improve treatment outcomes for this advanced disease. In conclusion, our data demonstrate the synergistic antitumor effects of CTLI against DTC cells and xenograft models. This potential combination therapy suggests a powerful and tolerable new treatment strategy for patients with advanced thyroid cancer.

## 4. Materials and Methods

### 4.1. Cell Culture

Human DTC cell lines, both K1 (papillary thyroid cancer, PTC cell line) and FTC-133 (follicular thyroid cancer, FTC cell line), were obtained from the European Collection of Authenticated Cell Cultures (Wiltshire, UK, EC92030501-F0, EC94060901-F0, respectively). K1-NIS cells were obtained by transfection using the human SLC5A5 (NIS) gene clone (OriGene) into K1 cells and were kindly donated by Dr. Watabe and Dr. Kaneda [29]. K1 and K1-NIS cells were cultured in a mixture of Dulbecco’s Modified Eagle Medium (DMEM; Sigma-Aldrich, St. Louis, MI, USA, D6429), Ham’s F12 medium (Sigma-Aldrich, N6658), and MCDB 105 medium (Sigma-Aldrich, M6395) (2:1:1, *v*/*v*/*v*) with 10% fetal bovine serum (FBS; Thermo Fisher Scientific, Inc., Walthma, MA, USA, 10270106). Stable cell lines expressing NIS gene were obtained by screening infected K1-NIS cells using G418 (Nacalai Tesque, Kyoto, Japan, 09380-44) at 500 µg/mL. FTC-133 cells were cultured in a mixture of DMEM and Ham’s F12 medium (1:1, *v*/*v*) with 10% FBS. All cells were grown at 5% CO_2_ and 37 °C. The media were replaced twice per week.

### 4.2. Western Blot Analysis

Collected cellular proteins from K1 or K1-NIS cells were electrophoresed on a sodium dodecyl sulfate-polyacrylamide gel (Mini-PROTEAN^®^ TGX^TM^ precast gel, Bio-Rad, Hercules, CA, USA, 4569034) and transferred to a polyvinylidene fluoride membrane (Bio-Rad, 1704156). The membranes were incubated with the following primary antibodies: NIS (Abcam, Cambridge, UK, ab83816) and β-actin (Cell Signaling, Beverly, MA, USA, 3700). Immune-reactive bands were detected by image scan using a LI-COR Odyssey Imaging System (LI-COR Biosciences, Lincoln, NE, USA).

### 4.3. Cell Proliferation Assay

K1 and K1-NIS cells (2 × 10^3^ cells/well) were seeded and cultured in 96-well culture plates and incubated for 0, 24, 48, 72, or 96 h, and the blank group contained only medium without cells. WST-8 (Dojindo, Kumamoto, Japan) at 10 μL was added to the wells, and the cells were incubated for 4 h at 5% CO_2_ and 37 °C. The absorbance was then measured at a wavelength of 450 nm and compared with a reference measurement at 630 nm using an iMark^TM^ microplate reader (Bio-Rad, Hercules, CA, USA). Each experiment was performed four times.

### 4.4. In Vitro ^125^I Accumulation Study

A modified protocol as previously described was used for iodide accumulation studies [34]. Briefly, K1, FTC-133, or K1-NIS cells treated with vehicle (0.1% DMSO in media) or 3 and 20 μM lenvatinib mesilate (lenvatinib; supplied from Eisai Co., Ltd., Tokyo, Japan) for 48 h were harvested using Accutase (Innovative Cell Technologies, Inc., California, USA, AT-104) and suspended in 7.0 mL of complete medium at a concentration of 1–2 × 10^6^ cells/mL. Cell suspensions were incubated at 37 °C in a stirred water bath. Samples (300 μL) were obtained in duplicate at 1, 15, 30, 45, and 60 min after adding 370 kBq of ^125^I (Perkin Elmer, Inc., Waltham, MA, USA, NEZ033) and transferred to 1.5 mL microcentrifuge tubes containing 1 mL ice-cold saline. These samples were then centrifuged at 14,000 rpm for 2 min to produce cell pellets, the supernatant was aspirated, and the remaining pellets were carefully washed with 0.5–1.0 mL of ice-cold saline to remove the remaining unbound ^125^I. The sample radioactivity was counted using an automatic γ counter (WIZARD “3” 1480; Perkin Elmer). The intracellular accumulation ratio (CPMin/CPMout) was calculated as the ratio of radioactivity concentration inside the cell to that found in the supernatant. The accumulation ratio equation was calculated as follows using the radioactivity measurements in cell pellets and a standard representing the supernatant concentration together with an independent cell volume measurement [34]:$$\mathrm{CPMin}/\mathrm{CPMout}=\frac{{\mathrm{counts}\;\mathrm{per}\;\mathrm{minute}}_{\mathrm{pellet}}\times \#{\mathrm{cells}}_{\mathrm{pellet}}}{{\mathrm{counts}\;\mathrm{per}\;\mathrm{minute}}_{\mathrm{supernatant}}}\;$$
where counts per minute_pellet_ is the measured radioactivity rate of the pellet, counts per minute_supernatant_ is the measured radioactivity rate of the supernatant, volume_pellet_ = volume_supernatant_ = 300 μL, and #cells_pellet_ = volume_pellet_ × density_pellet_ = 300 μL × (3.1, 2.8, or 3.3 × 10^5^ cells/μL)*, where * is the constant density per 1 μL of packed K1, FTC-133, or K1-NIS cells, respectively. The constant density of packed cells was measured using a Packed Cell Volume Tube (Techno Plastic Products AG, Trasadingen, Switzerland, 87005). These studies on ^125^I accumulation were repeated five times.

### 4.5. In Vivo ^123^I Uptake Experiments

All animal experiments were approved by the Animal Experiment Committee of Kansai Medical University (21-001), and all methods involving animals were conducted following the relevant guidelines and regulations. SCID beige mice (CB17.Cg-Prkdc^scid^ Lyst^bg-J^/CrlCrlj, female, 5 weeks old) were obtained from Charles River Laboratories Japan (Kanagawa, Japan). Mice were maintained under specific pathogen-free conditions and housed in barrier facilities on a 12 h light/dark cycle, with food and water ad libitum. K1-NIS cells were resuspended in PBS and Matrigel (Geltrex™ LDEV-Free Reduced Growth Factor Basement Membrane Matrix; Thermo Fisher Scientific, Inc., Waltham, MA, USA, A1413201) (1:1, *v*/*v*). Thereafter, 5 × 10^6^ cells were subcutaneously inoculated into the dorsal right shoulder. K1-NIS tumor xenograft mice (10 weeks old; body weight of 20.45 ± 0.96 g) were orally administered vehicle (sterile distilled water) or lenvatinib (10 mg/kg) once a day (*n* = 5 per group) for 5 days before imaging analysis with radioactive ^123^I 4 weeks after implantation, when the tumor size reached approximately 10 mm in diameter. A 10-day thyroxine treatment was initiated before the imaging experiment to suppress thyroid iodine uptake. Each day, animals were intraperitoneally (IP) injected with 2 μg of L-thyroxine (Sigma-Aldrich, T2376) diluted in 100 μL of PBS. Na^123^I (FUJIFILM RI Pharma Co., Ltd., FRI123I) at 3.8 MBq was orally administered in 200 μL of sterile saline for each mouse. Mice were anesthetized by isoflurane inhalation after Na^123^I administration. The planar imaging was performed over 10 min at each imaging time using an Inveon Multimodality system (Siemens Healthineers, Erlangen, Germany) equipped with a low-energy ultrahigh-resolution parallel collimator. The gamma ray energy photo peak was adjusted at 159 keV with a window of ±10%. The image acquisition of planar imaging was performed at 10 min, 1, 3, and 6 h, followed by an integrated single-photon emission computed tomography and computed tomography (SPECT/CT) imaging using a small animal micro-SPECT scanner. CT images were acquired before obtaining whole-body NanoSPECT images. The images were processed and reconstructed using the COBRA (Siemens Healthineers), IAW (Siemens Healthineers) for image acquisition, and Inveon Viewer software (Siemens Healthineers) for image analysis. Regions of interest (ROIs) were drawn around the tumor and thyroid, and the ROI radioactivity (ROI_tumor_ and ROI_thyroid_, respectively) was measured using the ASIPro software (Siemens Healthineers). The tumors were harvested and weighed after imaging analysis. The tumor radioactivity was counted using an automatic ɣ counter (WIZARD “3” 1480; Perkin Elmer).

### 4.6. Colony Formation Assay and Wound-Healing Assay

Cell suspensions of K1 and K1-NIS (1 × 10^6^ cells) were incubated with 200 μL medium with/without 740 kBq of Na^131^I (FUJIFILM RI Pharma Co., Ltd., 101-ZR034) in 1.5 mL microcentrifuge tubes for 30 min at 37 °C with 5% CO_2_. These samples were then centrifuged at 1500 rpm for 3 min to produce cell pellets, the supernatant was aspirated, and the remaining pellets were resuspended with 500 μL of PBS and centrifuged at 1500 rpm for 3 min to remove the remaining unbound ^131^I. The remaining pellets were resuspended with 200 μL of medium and incubated in 1.5 mL microcentrifuge tubes for 3 h at 37 °C with 5% CO_2_ after removing the supernatant. These cells were seeded in six-well plates (2 × 10^4^ cells per well) and allowed to attach overnight for the colony formation assay. The cells were then treated with vehicle (0.1% DMSO) or 20 µM of lenvatinib in culture medium for 6 days (7 days after Na^131^I treatment). Surviving colonies (≥50 cells per colony) were fixed with acetic acid and methanol (1:7, *v*/*v*), stained with 0.5% crystal violet, and counted.

Cells treated with Na^131^I (or control cells) were plated in 24-well plates (1 × 10^5^ cells per well) for wound-healing assay. The individual wells were wounded by scratching with a pipette tip and washed with medium after 48 h of incubation when cells reached 80% confluence, and cells were then treated with vehicle (0.1% DMSO) or 20 µM of lenvatinib in culture medium for 24 h at 37 °C with 5% CO_2_. Cells at the two time points (0 and 24 h) were photographed under phase contrast microscopy to compare the wound closure rate in each group. The wound area and the wound coverage of the total area were measured using ImageJ^®^ (Rasband, W.S., National Institutes of Health, Bethesda, MD, USA). The wound closure rate was calculated as follows [35]:$$\mathrm{Wound}\;\mathrm{closure}\;\mathrm{rate}\;(\%)=\left(\frac{A_{t=0}-{\;A}_{t=∆t}}{A_{t=0}}\right)\;\times \;100$$
where A_t=0_ is the initial wound area and A_t=__△t_ is the wound area after 24 h. Each experiment was performed at least three times.

### 4.7. In Vivo Combination Therapy with ^131^I and Lenvatinib

NOD SCID mice (NOD.CB17-Prkdcscid/J, male, 5 weeks old) were obtained from Charles River Laboratories Japan (Kanagawa, Japan). K1-NIS cells were resuspended in PBS and Matrigel (Geltrex LDEV-Free Reduced Growth Factor Basement Membrane Matrix) (1:1, *v*/*v*). Thereafter, 5 × 10^6^ cells were subcutaneously inoculated into the dorsal right shoulder. The tumor volumes of K1-NIS xenografts reached almost 300 mm^3^ 2 weeks after transplantation. The mice were then randomly divided into four groups: vehicle (sterile distilled water), ^131^I alone, lenvatinib alone, and combination with ^131^I and lenvatinib (*n* = 6 per group). Mice, in the ^131^I alone and combination groups, were intravenously (IV) injected with 3.7 MBq of Na^131^I (FUJIFILM RI Pharma Co., Ltd., Chiba, Japan, 101-ZR034) diluted in 100 μL of PBS through the tail vein on day 0. Mice, in the other two groups, were IV injected with 100 μL of PBS for vehicle control on day 0. Lenvatinib (10 mg/kg) was orally administered 3 h after ^131^I injection on day 0 and repeated once daily. The lenvatinib dose was decided based on a previous study [16,25]. The tumor sizes were measured every 2 days using a caliper, and tumor volumes were estimated using the following formula: tumor volume (mm^3^) = 1/2 L × S^2^ (L, longest diameter; S, shortest diameter). The change in tumor volume in the treated group relative to that of the control group was calculated according to the following formula: ΔT/C = (ΔT/ΔC) × 100%, where ΔT and ΔC are the tumor volume changes in the treated and vehicle control groups, respectively. Euthanasia was performed by cervical dislocation under deep anesthesia with medetomidine and ketamine. Subsequently, the tumors were harvested and weighed.

### 4.8. Statistical Analysis

Data are presented as means ± standard errors of the mean (SEMs). Statistical significance was evaluated using Student’s *t* test or ANOVA by the GraphPad Prism version 8 software (GraphPad, San Diego, CA, USA). A *p*-value of <0.05 was considered significant for all tests.

## Figures and Tables

**Figure 1 ijms-23-09872-f001:**
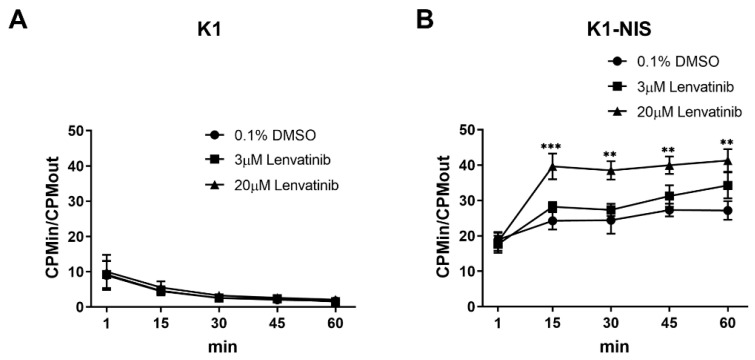
Increased intracellular uptake of radioiodine through sodium iodide symporter after lenvatinib treatment in vitro. (**A**,**B**) Intracellular accumulation ratio (CPMin/CPMout) of ^125^I in K1 and K1-NIS cells treated with 0.1% DMSO or 3 and 20 µM lenvatinib for 48 h. Cells at a concentration of 1–2 × 10^6^ cells/mL were obtained at 1, 15, 30, 45, and 60 min after adding 370 kBq of ^125^I, and the ratio of radioactivity concentration inside the cell to that found in the supernatant was calculated (*n* = 5). Data represent means ± SEM. ** *p* < 0.01, *** *p* < 0.001 vs. control.

**Figure 2 ijms-23-09872-f002:**
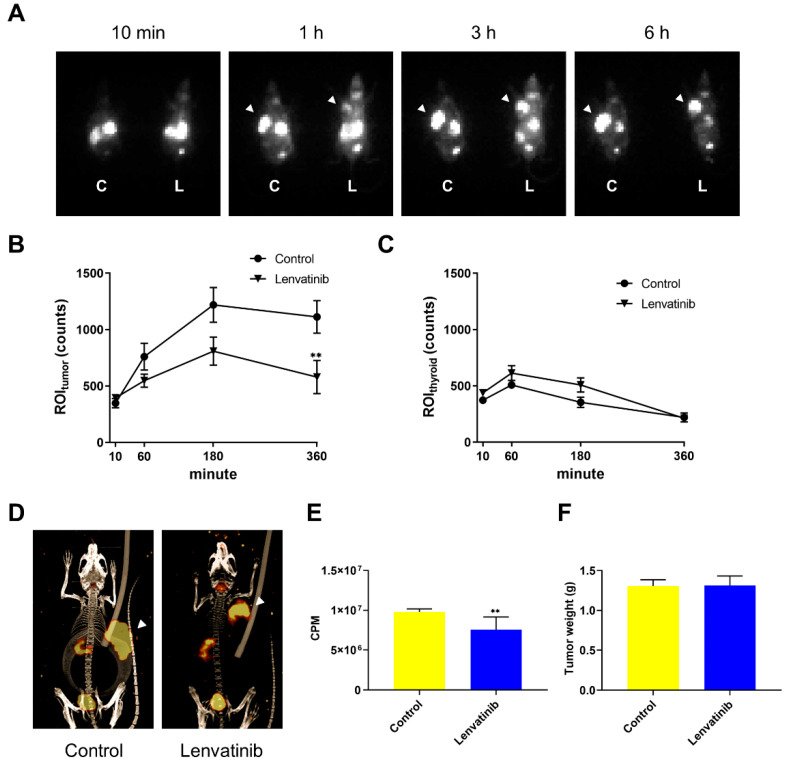
In vivo DTC radioiodine avidity after lenvatinib treatment. The representative planar images (**A**) and quantification of regions of interest (ROIs) within the tumor area (**B**) and thyroid (**C**). C, control; L, lenvatinib. Arrowheads indicate tumor area (**A**). (**D**) The representative images of SPECT/CT (left panel, control group; right panel, lenvatinib group). Arrowheads indicate tumor area. (**E**) The radioactivity of the counted tumor using an automatic ɣ counter. (**F**) Weights of the dissected xenograft tumors. K1-NIS cells at 5 × 10^6^ were subcutaneously transplanted in SCID beige mice. Mice were orally administered vehicle or lenvatinib (10 mg/kg) once a day (*n* = 5 per group) for 5 days before imaging analysis. The planar imaging was performed at 10 min, 1, 3, and 6 h after oral administration of 3.8 MBq of Na^123^I, followed by SPECT/CT imaging. ROIs were drawn around the tumor and thyroid, and the radioactivity in the ROIs was measured. Data represent means ± SEM. ** *p* < 0.01 vs. control.

**Figure 3 ijms-23-09872-f003:**
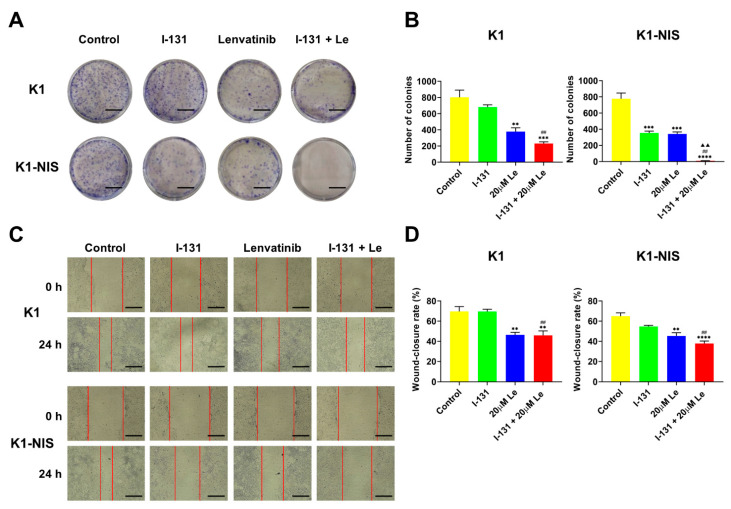
Thyroid cancer cell growth and migration inhibition using combination therapy with ^131^I and lenvatinib in vitro. Images (**A**) and quantification (**B**) of colony formation assay indicate K1 and K1-NIS cells treated with vehicle (0.1% DMSO) or 20 µM of lenvatinib in culture medium for 6 days following treatment with/without Na^131^I (7 days after treatment with Na^131^I) (*n* = 3). Scale bars, 10 mm (**A**). Images (**C**) and quantifications (**D**) of wound-healing assay indicate K1 and K1-NIS cells treated with vehicle (0.1% DMSO) or 20 µM lenvatinib in culture medium for 24 h after scratching following treatment with/without Na^131^I (3 days after treatment with Na^131^I) (*n* = 4). The red lines define the area lacking cells. Scale bars, 1000 µm (**C**). Data represent means ± SEM. ** *p* < 0.01, *** *p* < 0.001, **** *p* < 0.0001 vs. control; ^##^
*p* < 0.01 vs. ^131^I monotherapy; ^▲▲^
*p* < 0.01 vs. lenvatinib monotherapy. Le, lenvatinib.

**Figure 4 ijms-23-09872-f004:**
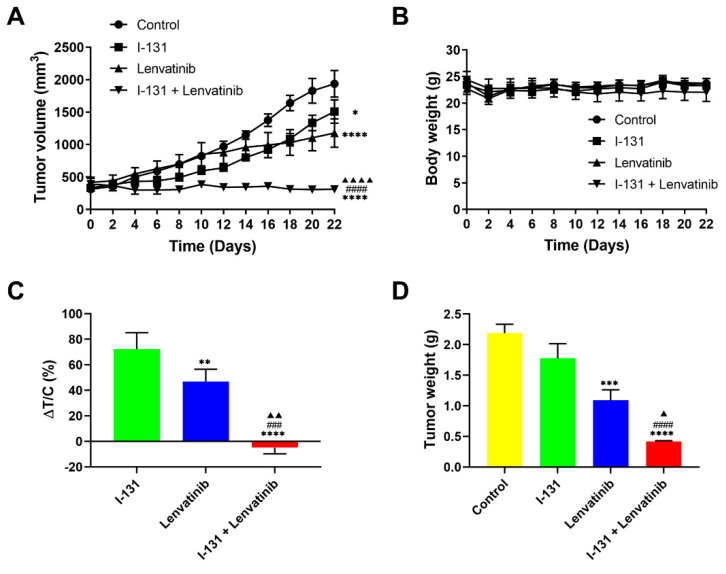
Synergistic anticancer effects by combination therapy in thyroid cancer xenografts. (**A**,**B**) Time course of xenograft tumor growth and animal weight in each group. (**C**) Change in tumor volume in the treated group relative to that in the control group. ΔT/C (%) was calculated as (ΔT/ΔC) × 100%, where ΔT and ΔC are the tumor volume changes for the treated and vehicle control groups, respectively. (**D**) Weights of the dissected xenograft tumors. K1-NIS cells at 5 × 10^6^ were subcutaneously transplanted in NOD SCID mice, and mice were intravenously injected with vehicle (PBS) or 3.7 MBq of Na^131^I via the tail vein. Lenvatinib (10 mg/kg) or vehicle (sterile distilled water) was orally administered 3 h after ^131^I injection and repeated once daily. Data represent means ± SEM. (*n* = 6/group). * *p* < 0.05, ** *p* < 0.01, *** *p* < 0.001, **** *p* < 0.0001 vs. control; ^###^
*p* < 0.001, ^####^
*p* < 0.0001 vs. ^131^I monotherapy; ^▲^
*p* < 0.05, ^▲▲^
*p* < 0.01, ^▲▲▲▲^
*p* < 0.0001 vs. lenvatinib monotherapy. Le, lenvatinib.

## Data Availability

Not applicable.

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
