# Peer review of "Efficacy of Combination Therapy with Lenvatinib and Radioactive Iodine in Thyroid Cancer Preclinical Model"

_ijms, 2022, doi:10.3390/ijms23179872_

Round 1

Reviewer 1 Report

In the present report Suzuki investigated the combination therapy with lenvatinib and radioactive iodine treatment. The results showed that administration of lenvatinib after radioactive iodine effectively curtailed the progression of thyroid cancer both in vivo and in vivo models. The methods used in this study are well suited for this kind of investigations and the results are easy to follow. I have a few comments that should be considered.

Line 84: “…into K1 cells and were kindly divided by Dr. Watabe and..” I think the word “donated” would be better than “divided”.

Line 195, please write 131 as superscript.

Figure 1: please make the figure a bit bigger or increase the size of the symbols. Now it is very difficult to actually see the results clearly.

Lines 255-257: The speculation regarding the effect of lenvatinib on angiogenesis should be removed to the discussion.

Figure 2B,C: as with Fig 1, the symbols are very small and difficult to read.

Figure 3D: is there a difference between K1 and Ki-NIS in the groups treated with 1-131 + lenvatinib? This might be important when discussing the efficacy of the treatments between the groups.

Figure 4A,B: as previously mentioned regarding the size of the symbols.

Line 337: do the Authors have any ideas on what the mechanism might be regarding the effect of lenvatinib and the enhanced uptake of radioiodine?

Author Response

Dear Reviewer:

Thank you very much for reviewing our manuscript and offering valuable advice. We have addressed your comments with point-by-point responses, and revised the manuscript accordingly.

Manuscript ID: ijms-1872128

Title: Efficacy of Combination Therapy with Lenvatinib and Radioactive Iodine in Thyroid Cancer Preclinical Model

Reviewer’s Comments:
In the present report Suzuki investigated the combination therapy with lenvatinib and radioactive iodine treatment. The results showed that administration of lenvatinib after radioactive iodine effectively curtailed the progression of thyroid cancer both in vivo and in vivo models. The methods used in this study are well suited for this kind of investigations and the results are easy to follow. I have a few comments that should be considered.

Line 84: “…into K1 cells and were kindly divided by Dr. Watabe and..” I think the word “donated” would be better than “divided”.

Line 195, please write 131 as superscript.

Figure 1: please make the figure a bit bigger or increase the size of the symbols. Now it is very difficult to actually see the results clearly.

Lines 255-257: The speculation regarding the effect of lenvatinib on angiogenesis should be removed to the discussion.

Figure 2B,C: as with Fig 1, the symbols are very small and difficult to read.

Figure 3D: is there a difference between K1 and Ki-NIS in the groups treated with 1-131 + lenvatinib? This might be important when discussing the efficacy of the treatments between the groups.

Figure 4A,B: as previously mentioned regarding the size of the symbols.

Line 337: do the Authors have any ideas on what the mechanism might be regarding the effect of lenvatinib and the enhanced uptake of radioiodine?

Response to the Reviewer’s comment:
Thank you for providing these insights. These comments have helped us significantly improve the manuscript. We have answered each of your points below.

Comment 1:
Line 84: “…into K1 cells and were kindly divided by Dr. Watabe and..” I think the word “donated” would be better than “divided”.

Response: Thank you for your suggestion. We have revised the text as follows (Line 84):
(Before the change)
…into K1 cells and were kindly divided by Dr. Watabe and...
(After the change)
…into K1 cells and were kindly donated by Dr. Watabe and...

Comment 2:
Line 195, please write 131 as superscript.

Response: Thank you for your suggestion. We have revised the text as follows (Line 195):
(Before the change)
2.7 In vivo combination therapy with 131I and lenvatinib
(After the change)
2.7 In vivo combination therapy with 131I and lenvatinib

Comment 3:
Figure 1: please make the figure a bit bigger or increase the size of the symbols. Now it is very difficult to actually see the results clearly.
Response: Thank you for your suggestion. We have increased the size of Figure 1 and its symbols. We have also increased the size of the symbols of Fig. S1B and S2.

Comment 4:
Lines 255-257: The speculation regarding the effect of lenvatinib on angiogenesis should be removed to the discussion.
Response: Thank you for your suggestion. We have removed the following sentence (Lines 255-257):
“These results suggest that lenvatinib administration before 131I treatment potentially induces a decreased radioiodine avidity through tumor angiogenesis inhibition.”

We have revised the following text in the discussion (Lines 340-341):
(Before the change)
…, suggesting that tumor angiogenesis inhibition affects the radioiodine avidity.
(After the change)
…, suggesting that tumor angiogenesis inhibition of lenvatinib affects the radioiodine avidity.

Comment 5:
Figure 2B,C: as with Fig 1, the symbols are very small and difficult to read.
Response: Thank you for your suggestion. We have increased the size of Figure 2 and the symbols of Figure 2B,C.

Comment 6:
Figure 3D: is there a difference between K1 and Ki-NIS in the groups treated with 1-131 + lenvatinib? This might be important when discussing the efficacy of the treatments between the groups.

Response: Thank you for your suggestion. For wound-healing assay, we have analyzed the difference between K1 and K1-NIS in the groups treated with 1-131 + lenvatinib. The inhibitory effect of CTLI was tended to be higher in K1-NIS cells than in K1 cells, although it was not significant difference. We have revised the text as follows (Lines 278-281):
(Before the change)
Results showed that CTLI significantly inhibited migration ability in K1-NIS cells (Fig. 3C, D), thereby suggesting a synergistically inhibited proliferation and migration ability of thyroid cancer cells, especially NIS-expressing DTC cells in vitro, with the combined therapy.

(After the change)
Results showed that CTLI significantly inhibited migration ability in both cell lines, and the inhibitory effect was tended to be higher in K1-NIS cells than in K1 cells (Fig. 3C, D). These results suggest that the combined therapy synergistically inhibited proliferation and migration ability of thyroid cancer cells, especially NIS-expressing DTC cells in vitro.

Comment 7:
Figure 4A,B: as previously mentioned regarding the size of the symbols.
Response: Thank you for your suggestion. We have increased the size of Figure 4 and the symbols of Figure 4A,B.

Comment 8:
Line 337: do the Authors have any ideas on what the mechanism might be regarding the effect of lenvatinib and the enhanced uptake of radioiodine?

Response:
Thank you for your comment. As you mentioned, we could not reveal the mechanism of the increased iodine uptake after lenvatinib treatment in vitro. Therefore, we didn’t mention about this issue in our manuscript. Further investigations are needed to elucidate the radioiodine kinetics and its mechanism after lenvatinib administration in vitro and in vivo.

Again, we would like to thank you for providing us the opportunity to strengthen our manuscript with your valuable comments and queries. We have worked hard to incorporate your feedback and hope that these revisions persuade you to accept our manuscript for publication.

Sincerely,

Kensuke Suzuki
Department of Otolaryngology, Head and Neck Surgery, Kansai Medical University
Shinmachi 2-5-1, Hirakata, Osaka 573-1010, Japan
Phone No: +81 72 804 0101
Fax No: +81 72 804 2547
Email Address: [email protected]

Reviewer 2 Report

This original paper by Suzuki and colleagues is facing a very contemporary issue of the treatment of thyroid cancer: the finding of new therapeutic strategy to fight the small percent of aggressive forms of this disease.

Overall the study is well designed and well presented, and it could represent a good model of fine-developed translational science experiment. However, the reported results are in part discordant between in vitro and in vivo settings , in particular, regarding the timing of the Lenvatinib administration (see lines 335-346). (What could be the explanation of this effect?)

Moreover the experiment is based on the anticipated knowledge of the "NIS status" of the examined cancer cells, which is quite impossible to obtain in patients (but only deducible after different therapeutic attempts). For this reason, I find difficult to apply the author's conclusions to real world. 

Said that, the design of this research could  improve our knowledge and expertise on thyroid cancer preclinical studies.

Author Response

Dear Reviewer:

Thank you very much for reviewing our manuscript and providing valuable comments. We have addressed your comments with point-by-point responses, and revised the manuscript accordingly.

Manuscript ID: ijms-1872128

Title: Efficacy of Combination Therapy with Lenvatinib and Radioactive Iodine in Thyroid Cancer Preclinical Model

Reviewer’s Comments:
This original paper by Suzuki and colleagues is facing a very contemporary issue of the treatment of thyroid cancer: the finding of new therapeutic strategy to fight the small percent of aggressive forms of this disease.
Overall the study is well designed and well presented, and it could represent a good model of fine-developed translational science experiment. However, the reported results are in part discordant between in vitro and in vivo settings , in particular, regarding the timing of the Lenvatinib administration (see lines 335-346). (What could be the explanation of this effect?)
Moreover the experiment is based on the anticipated knowledge of the "NIS status" of the examined cancer cells, which is quite impossible to obtain in patients (but only deducible after different therapeutic attempts). For this reason, I find difficult to apply the author's conclusions to real world. 
Said that, the design of this research could  improve our knowledge and expertise on thyroid cancer preclinical studies.

Response to the Reviewer’s comment:
Thank you for providing these insights. We have answered each of your points below.

Comment 1:
However, the reported results are in part discordant between in vitro and in vivo settings , in particular, regarding the timing of the Lenvatinib administration (see lines 335-346). (What could be the explanation of this effect?)

Response: Thank you for your comment. We have set the timing of lenvatinib administration for in vitro 125I accumulation study and in vivo 123I uptake experiments based on the results of our preliminary experiments. As shown in the results (Fig.2B), 123I uptake into the tumor peaked at 3 h after 123I administration. Since the pharmacokinetics of 123I and 131I are presumed to be similar, lenvatinib treatment was started at the time of 131I uptake by the tumor (3h after administration) in the combination therapy with 131I and lenvatinib in vivo (lines 343-345).

Comment 2:
Moreover the experiment is based on the anticipated knowledge of the "NIS status" of the examined cancer cells, which is quite impossible to obtain in patients (but only deducible after different therapeutic attempts). For this reason, I find difficult to apply the author's conclusions to real world. 

Response: Thank you for your comment. As you mentioned, it is difficult to achieve stable NIS function in patients, and it is unclear whether the effects of CTLI in this study can be replicated in real-world practice. We have added the following text in Discussion for future perspective (lines 364-366), leading to the topic of MAPK inhibitors, which is discussed in the next paragraph:
The use of agents that enhance NIS function and RAI uptake in combination with CTLI may be necessary for the clinical application.

Again, we would like to thank you for providing us the opportunity to strengthen our manuscript with your valuable comments and queries. We have worked hard to incorporate your feedback and hope that these revisions persuade you to accept our manuscript for publication.

Sincerely,

Kensuke Suzuki
Department of Otolaryngology, Head and Neck Surgery, Kansai Medical University
Shinmachi 2-5-1, Hirakata, Osaka 573-1010, Japan
Phone No: +81 72 804 0101
Fax No: +81 72 804 2547
Email Address: [email protected]